# Dietary Counseling Aimed at Reducing Sugar Intake Yields the Greatest Improvement in Management of Weight and Metabolic Dysfunction in Children with Obesity

**DOI:** 10.3390/nu14071500

**Published:** 2022-04-03

**Authors:** Aurelia Radulescu, Mary Killian, Qiwen Kang, Qingcong Yuan, Samir Softic

**Affiliations:** 1Department of Pediatrics, Kentucky Children’s Hospital, University of Kentucky College of Medicine, Lexington, KY 40536, USA; aurelia.radulescu@uky.edu; 2Department of Pediatrics, University of Kentucky College of Medicine, Lexington, KY 40536, USA; mary.killian@vumc.org; 3Department of Pediatrics, Vanderbilt University Medical Center, Nashville, TN 37232, USA; 4Department of Statistics, College of Arts and Sciences, University of Kentucky, Lexington, KY 40536, USA; qiwen.kx@gmail.com (Q.K.); qingcong.yuan@miamioh.edu (Q.Y.); 5Department of Statistics, Miami University, Oxford, OH 45056, USA; 6Department of Pharmacology and Nutritional Sciences, University of Kentucky College of Medicine, Lexington, KY 40536, USA; 7Section on Integrative Physiology and Metabolism, Joslin Diabetes Center, Harvard Medical School, Boston, MA 02115, USA

**Keywords:** sugar, dietary counseling, pediatric, obesity, metabolic dysfunction

## Abstract

Pediatric obesity is a significant public health problem, the negative outcomes of which will challenge individual well-being and societal resources for decades to come. The objective of this study was to determine the effects of dietary counseling on weight management and metabolic abnormalities in children with obesity. One hundred and sixty-five patients aged 2–18 years old were studied over a two and a half year period. Data collected included demographic information, anthropometric assessment, laboratory measurements, and self-reported eating behaviors. Dietary counseling was provided at each visit. The data was analyzed from the first and last visits and the subjects were retrospectively divided into responders and non-responders based on a decrease in their BMI. After receiving dietary guidance, BMI decreased in 44% of the children, and these participants were classified as responders (BMI-R; *n* = 72). However, BMI did not improve in 56% of the participants, and these were classified as non-responders (BMI-NR; *n* = 93). At the initial visit, anthropometric measurements and dietary habits were similar between the groups. At the time of the last visit, mean change in BMI was −1.47 (SD 1.31) for BMI-R and +2.40 (SD 9.79) for BMI-NR. Analysis of food intake revealed that BMI-R significantly improved their dietary habits (*p* = 0.002) by reducing the intake of sugar-sweetened beverages (*p* = 0.019), processed foods (*p* = 0.002), sweets (*p* < 0.001), and unhealthy snacks (*p* = 0.009), as compared with BMI-NR. There was no change in the intake of second helpings, portion sizes, skipping meals, frequency of meals eaten at school, condiment use, intake of fruits and vegetables and consumption of whole grains between the groups. BMI-R also achieved an improvement in fasted glucose (*p* = 0.021), triglycerides (*p* < 0.001), and total cholesterol (*p* = 0.023), as compared to BMI-NR. In conclusion, children with obesity who were able to decrease their BMI implemented a significant reduction in consumption of foods with high sugar content. Focusing on reducing sugar intake may yield the biggest impact in terms of weight management and the improvement of metabolic abnormalities.

## 1. Introduction

The prevalence of pediatric obesity has significantly increased in the last few decades, and represents one of the most common chronic disorders, affecting an estimated 14.4 million children and adolescents in the USA [1]. The new evidence suggests that this problem has worsened during the COVID-19 pandemic, affecting 19.3% children in 2019, and 22.0% in 2021, respectively [2]. Higher prevalence and the severity of obesity in children leads to a greater risk of developing obesity-associated complications including dyslipidemia, hypertension, type 2 diabetes mellitus, nonalcoholic fatty liver disease, obstructive sleep apnea, and others, which traditionally used to develop much later in adulthood [3,4]. The data from our clinic suggest that 62.2% of boys and 56.8% of girls over ten years of age with severe obesity have already developed three or more components of metabolic dysfunction [5].

The etiology of childhood obesity is multifactorial, but at the core of the issue is a positive energy balance produced by increased caloric intake and decreased energy expenditure. Intake of a high-fat diet has long been suspected to drive the obesity epidemic. As a result of intense research, public awareness, and policy changes, total intake of fat has stabilized, and the percentage of calories ingested from saturated fat has decreased over the last several decades [6]. Unfortunately, curbed fat intake did not translate into a substantial decrease in obesity prevalence. This has resulted in a paradigm shift and more attention is now placed on reducing the intake of dietary sugar. Table sugar, sucrose, is a disaccharide composed of glucose and fructose monomers. Fructose intake, in particular, has been associated with the development of obesity and metabolic dysfunction. Several medical societies recommend decreasing fructose intake as a way to manage obesity and its complications. Furthermore, a number of leading pharmaceutical companies are developing inhibitors of fructose metabolism as a treatment for obesity, insulin resistance, and non-alcoholic fatty liver disease [7,8].

While effective treatment options for children with obesity are being developed, intensive lifestyle modifications are strongly recommended for the management of children with obesity [9]. Based on current recommendations, the age-based appropriate lifestyle changes should include dietary modifications, increasing physical activity, and decreasing non-school related screen time [10]. The emphasis on dietary counseling is to improve the quality of food, rather than to restrict caloric intake, in order to protect normal growth and development [11,12]. Examples of this approach include the stoplight diet [13], the US Department of Agriculture Food Guide Pyramid [14] and the American Heart Association guidelines for the prevention of cardiovascular disease beginning in childhood [15]. However, what dietary changes result in the greatest improvement in pediatric weight management is still an area of ongoing research.

This study was set up to evaluate the changes implemented by patients following intensive dietary counseling based on US Department of Agriculture Food and Nutrition Service MyPlate guidelines [16]. This is a retrospective data analysis of the patients that were successful in reducing their BMI with this guidance, versus the ones that had no improvement in their BMI. We hypothesized that the difference in food intake between BMI responders and non-responders might uncover dietary interventions that should be prioritized during a busy general pediatric clinic visit, in order to yield the largest benefit in terms of weight management and the improvement of metabolic dysfunction.

## 2. Methods

This study is a retrospective review of medical records of children between 2–18 years of age who were evaluated and treated for obesity (BMI ≥ 95th percentile for age and gender) at a regional weight management clinic over a two and a half year period. The study was approved by the University of Kentucky Medical Institutional Review Board, approval code 14-0504-P3H, and approval date 23 July 2017. All patient information was de-identified during the data collection process. For children who had two or more visits, the lab work completed at the first and last visits, and whose quantification of food intake and exercise was documented, were included in the study (*n* = 165). Patients with incomplete data, those receiving pharmacotherapy that could affect metabolic profile, and children with genetic abnormalities were excluded. The clinic’s personnel included an obesity medicine pediatric specialist, a registered dietician, and a clinical medical assistant.

At the initial visit, each child underwent a thorough history, physical examination, and laboratory screening. Data collected and analyzed included dietary pattern, demographic information, anthropometric measurements, parental weight status, and fasting labs including blood glucose, hemoglobin A1c (HbA1c), lipid panel, and liver enzymes. Weight, height, BMI, and manual blood pressure were measured at each visit. Follow-up visits were scheduled on average every two months but varied at the discretion of the provider from two weeks to six months. Most patients included in the study had seven visits. Laboratory tests were repeated annually unless a previous value had been abnormal, or if the child had gained significant amount of weight, in which case they were repeated sooner.

At each visit, a complete review of the child’s self-reported eating and exercise behaviors was conducted. This included frequency of second helpings, portion size, intake of sugar-sweetened beverages, consumption of processed foods, sweets, unhealthy snacks, skipping meals, frequency of meals eaten at school, condiment use, intake of fruits and vegetables, and servings of whole grains. For data analysis, food intake was scored based on the frequency of occurrence on a scale from 1 to 4, with a lower score corresponding to a healthier behavior (Appendix A). The sum of scored food intake behavior was termed the health behavior score (HBS) and was used to analyze the child’s food trends over time. We collected data on 11 food categories and dietary behaviors. The maximum score for each category was four, so the highest cumulative HBS of 44 indicates the worst diet. Data on exercise was collected as sessions per week and length of each session. A registered dietician and a pediatrician specializing in obesity medicine both counseled the family on appropriate healthy lifestyle changes. Dietary recommendations were based on US Department of Agriculture Food and Nutrition Service MyPlate guidelines [16]. Examples of counseling included eliminating additional servings, age-appropriate portions with balanced macronutrients, eliminating sugar-sweetened beverages, decreasing intake of processed foods and fast food, limiting sweets and unhealthy snacks, choosing healthy options at school meals, and increasing servings of fruits, vegetables, and whole grains. Recipes with healthy substitutions, ideas for packing breakfast and lunch from home, and healthy snack options were provided by the dietician. Exercise counseling included limiting non-school related screen time and sedentary activity to less than two hours per day, and increasing physical activity to one hour daily, as per expert committee recommendations [9]. The caregiver(s) and the patients were involved in these discussions at each visit. Any barriers to making the suggested changes were discussed to ensure a realistic plan was in place for the patient to follow.

### Statistical Analysis

Based on an absolute change in BMI of ≥0.10 from the first to last visit, patients were categorized into the following two groups: BMI responders (BMI-R) and BMI non-responders (BMI-NR). To analyze the difference in demographics and the biomedical profiles of patients in these two groups, we used Fisher’s exact tests, independent t tests, and chi-square tests. We calculated frequency tables for demographic and biomarker variables along with *p*-values for chi-square tests. The estimated mean and standard deviation, along with *p*-values for independent t tests, are also presented. Since patients varied in the number of total visits over the study period, we focused the analysis on their initial and final visits, as well as the interval change. We analyzed the relationship of parental obesity with the child’s BMI trend. To determine the changes in HBS over multiple visits, we compared BMI-R and BMI-NR using a variance-covariance matrix.

## 3. Results

### 3.1. The Effect of Dietary Counseling on BMI

The characteristics of the study population are shown in Appendix A. Of the subjects involved in the study, 43% were female, and 57% were male. About 45% were white, 28% were black, 24% were Hispanic, and 3% were biracial. We analyzed parental BMI from self-reported weight and height data obtained at the initial visit. For 78 patients, anthropometric data was available on both parents, including 99 patients with data on mothers and 85 with data on fathers. Among the available data, 62.6% of subjects had maternal BMI in the obese range, and 63.5% had fathers that were obese. The majority of children (71.8%) had both parents with BMI in either the overweight (BMI ≥ 25.0) or obese ranges. The prevalence of maternal obesity was significantly lower in BMI-R (48.7%), compared with 71.7% in BMI-NR (*p* = 0.021). Paternal obesity was similar between the two groups.

At the initial visit, the mean BMI was 31.49 Kg/m^2^, corresponding to the 98th percentile on the CDC growth charts (Table 1). Mean age, weight, height, BMI, and BMI percentile were similar between the groups. At the final visit, the mean BMI for the total study population was 32.20 Kg/m^2^, representing an increase of 0.712 (SD 7.63). There were no statistically significant differences between the groups in terms of weight, height, BMI, and BMI percentile at the end of the study. Age of the participants increased as expected (*p* = 0.04). Next, we identified the patients who were able to decrease their BMI and compared them to the subjects whose BMI continued to increase. Out of a total of 165 patients, 72 subjects (44%) improved their BMI and were classified as responders (BMI-R), while 93 (56%) did not reduce their BMI and were classified as non-responders (BMI-NR). There were no differences in age, weight, and height between responders and non-responders (Table 1). However, non-responders had a +2.40 (SD 9.79) unit increase in BMI, while responders decreased their BMI by −1.47 (SD 1.31) units (*p* < 0.001). Both groups had a decrease in their BMI percentile, but the BMI-R group had a significantly greater reduction of 1.0%, compared to a 0.2% reduction in the BMI-NR group (*p* = 0.011).

### 3.2. Dietary Changes Implemented by BMI Responders and Non-Responders

Dietary habits were compiled into a composite health behavior score (HBS), consisting of 11 dietary categories, and a numerical score was assigned to each category based on the frequency of occurrence, as outlined in Appendix A. At the initial visit, HBS was similar between the groups, at 36.4 ± 1.2 for BMI-R and 36.5 ± 1.3 for BMI-NR (Figure 1). Both groups demonstrated an improvement in HBS at the second visit, and this difference was sustained until the final visit. At the time of the final visit, BMI-R achieved a significantly lower HBS than the BMI-NR (*p* = 0.002), so that BMI-R had HBS of 25.8 ± 3.5 and BMI-NR of 30.2 ± 3.2, respectively. Next, we evaluated the impact of dietary counseling on individual dietary categories (Table 2). BMI-R showed a significant reduction in the intake of sugar-sweetened beverages (*p* = 0.019), processed foods (*p* = 0.002), sweets (*p* < 0.001), and unhealthy snacks (*p* = 0.009) when compared with BMI-NR. There was no difference between the groups in the frequency of second helpings, portion sizes, skipping meals, meals eaten at school, condiment use, intake of fruits and vegetables, and consumption of whole grains.

### 3.3. Metabolic Profile of BMI Responders versus Non-Responders

At the time of the initial visit, the patients had similar mean HbA1c, fasting blood glucose, serum ALT, and triglycerides (Table 3). Interestingly, BMI-R had higher total cholesterol (*p* = 0.02), LDL-c (*p* = 0.05), and HDL-c (*p* = 0.02) than BMI-NR. At the last visit, there was no difference between responders and non-responders in terms of HbA1c and serum ALT. The BMI-R group showed a strong tendency to have reduced fasting glucose (*p* = 0.07), compared to BMI-NR. Serum triglycerides were significantly lower in BMI-R (*p* = 0.01). Although the total cholesterol was significantly higher at the initial visit, it markedly decreased in BMI-R so that it was not different than in BMI-NR at the end of the study. Similarly, BMI-R had a significant decrease in HDL-c (*p* = 0.04), and a trend towards reduced LDL-c (*p* = 0.10), at the time of the last visit. When the total study population was analyzed together, in spite an overall increase in BMI, there was an improvement in serum triglycerides and total cholesterol. This difference is primarily driven by BMI responders who achieved significantly lowered triglycerides (*p* = 0.001), total cholesterol (*p* = 0.023), and also fasting glucose (*p* = 0.021), compared to BMI-NR.

## 4. Discussion

In the present study, we have retrospectively evaluated the impact of dietary counseling on food intake, anthropometric measurements, and serum markers of metabolic health. All the patients that received intensive dietary counseling implemented measurable changes in terms of improving the quality of their nutritional intake. For some patients, this resulted in decreased body mass index, and they were termed BMI responders. Others did not achieve a decrease in BMI and, as such, were classified as BMI non-responders. When evaluating the difference between BMI responders and non-responders, the former group achieved a significantly greater decrease in the intake of foods with high sugar content, such as sugar-sweetened beverages, sweets, processed food, and unhealthy snacks. There was no statistically significant difference between the groups in terms of the intake of second helpings, portion sizes, skipping meals, meals eaten at school, or the intake of whole grains, fruits, and vegetables. Compared to BMI non-responders, responders achieved a significant improvement in blood glucose, serum triglycerides, and total cholesterol. Our study suggests that focusing dietary counseling on reducing the intake of foods with high-sugar content may yield the greatest benefit in terms of improving BMI, as well as decreasing some measures of metabolic dysfunction.

High intake of dietary sugar has been linked with the development of obesity [17,18], insulin resistance [19,20], dyslipidemia [21,22], NAFLD [22,23,24], and many other metabolic abnormalities [20,25,26]. Several hypotheses have been proposed to explain this association. The first claims that sugar intake serves as a vehicle for increased caloric intake. Sugary deserts are traditionally consumed after a meal when the subject is no longer hungry. This translates into a higher caloric intake beyond the level determined by the homeostatic hunger cues. Similarly, sugar-sweetened beverages, which represent the largest means of sugar intake, are consumed when the subject is thirsty, not hungry, again contributing to increased caloric load. The second hypothesis proposes that the fructose component of dietary sugar is intrinsically harmful to cellular metabolic balance independent of total caloric intake. This is based on the studies in humans [27] and rodents [28], showing that fructose, but not equicaloric glucose intake, leads to development of metabolic complications. The pathways affected by fructose metabolism include insulin resistance [29], decreased fatty acid oxidation [30,31], and increased hepatic de novo lipogenesis [32,33], all of which contribute to fat accumulation. Mechanistically, these processes lead to inflammation [34], lipotoxicity [35], endoplasmic reticulum stress [36], advanced glycation end products [37], and others, which have been proposed to be the mediators of metabolic complications. On the other hand, isocaloric fructose restriction reduces insulin resistance [38], NAFLD [39], and low-density lipoprotein cholesterol [40,41] in human studies. Indeed, the World Health Organization [42] and several leading medical societies, such as the American Heart Association [43], the Canadian Diabetes Association [44], and the joint statement by the European Association for the Study of the Liver (EASL), the European Association for the Study of Diabetes (EASD), and the European Association for the Study of Obesity (EASO) [45], recommend reducing dietary sugar as a way to combat obesity and metabolic dysfunction.

While reducing sugar intake may explain the observed benefits in our responders versus non-responders, we cannot discount the possibility that additional factors may have contributed to this effect. BMI responders had a higher degree of dyslipidemia at the time of the initial visit. The presence of one or more metabolic complications may serve as an additional motivator to more strictly adhere to dietary recommendations [46]. Furthermore, BMI responders had a lower percentage of mothers with obesity. Parental obesity has been shown to greatly influence the weight status of children [47,48,49]. This, in part, reflects the shared family environment, parental awareness of healthy eating habits, and s similar genetic predisposition to obesity. Our study and others [50,51] indicate that maternal obesity is more strongly associated with childhood obesity than paternal weight status. This may be because the intrauterine environment is solely dictated by the mother. Conditions such as gestational diabetes, hypertension, pre-eclampsia, and placental insufficiency affect intrauterine growth, which can influence a long-term predisposition to the development of obesity [52,53]. Our dietary recommendations reflected the American Academy of Pediatrics [9] and Endocrine Society [4] guidance for the management of obesity in a pediatric population. Thus, our dieticians routinely counseled the patients on reducing their intake of high-fat foods. Unfortunately, we did not systemically collect data on reducing the intake of foods high in fat. However, we did collect data on adherence to our recommendations to reduce the quantity of food consumed, such as decreasing portions size and second helping. Additionally, we quantified the frequency of skipping meals and meals eaten at school, although we did not find a difference between the groups. Moreover, there was no difference in the intake of healthy foods such as fruits, vegetables, and whole grains. We did find a difference in reducing sugar intake, indicating that this intervention may be the easiest for the families to follow. Thus, in a busy general pediatric clinic, focused dietary counseling on reducing sugar intake may yield the best results in terms of improving BMI and metabolic dysfunction.

Our study is in line with other interventions that emphasize improving the quality of nutrition rather than solely advocating reduced caloric intake for the management of pediatric obesity. Hypocaloric diets are generally not recommended for children due to concerns regarding their safety, as well as their effects on normal growth and development [54]. Similar results to ours were demonstrated in a 12 week intervention study designed to reduce dietary fructose intake in children aged between five and eight years [55]. A reduction in weight in these subjects may be independent of the total caloric intake, as isocaloric fructose restriction in children has been shown not only to reduce weight, but also serum triglycerides, LDL cholesterol, and glucose tolerance [38,39,41]. Interestingly, we did not observe a significant reduction in serum ALT, which is often used as a surrogate marker to screen for fatty liver disease in children [56]. Similarly, AST was not different between responders and non-responders (*p* = 0.21). A recent study in children examined the effects of eight weeks of dietary sugar restriction and documented a reduction in ALT, as well as a decrease in hepatic de novo lipogenesis [33]. The difference between this study and ours may be that we did not specifically focus our dietary guidance on restricting fructose intake. Interventions designed to prioritize reducing sugar intake may yield greater results.

In summary, our study demonstrates that intensive dietary recommendations result in a meaningful and sustainable improvement in nutritional quality. Children with obesity who reduce their intake of foods high in sugar are able to achieve a statistically significant reduction in BMI and some measures of metabolic dysfunction. In a general pediatric practice, focusing on reducing the intake of dietary sugar may be the easiest strategy to implement in order to improve weight status and metabolic abnormalities in children with obesity.

## Figures and Tables

**Figure 1 nutrients-14-01500-f001:**
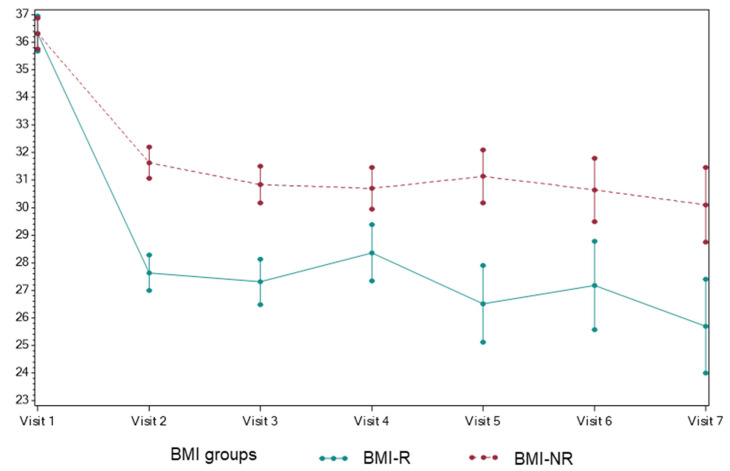
Health Behavior Score Trend over Time.

**Table 1 nutrients-14-01500-t001:** Treatment Effect by Group.

	Initial Visit				Last Visit				Interval Change			
	Total	BMI-NR	BMI- R	*p*-Value	Total	BMI-NR	BMI- R	*p*-Value	*p*- Total	Mean ∆, BMI-NR	Mean∆, BMI-R	*p*-Value
Age, y	10.16 (3.54)	10.05 (3.26)	10.32 (3.88)	0.62	10.97 (3.57)	10.99 (3.19)	10.94 (4.02)	0.93	0.04	0.94 (4.56)	0.62 (5.58)	0.686
Weight, kg	71.78 (34.18)	70.31 (34.58)	73.67 (33.79)	0.53	74.55 (31.82)	74.82 (29.48)	74.21 (34.82)	0.90	0.11	4.50 (18.88)	0.54 (12.38)	0.107
Height, cm	145.02 (21.33)	145.19 (19.10)	144.80 (24.04)	0.91	149.51 (19.16)	150.55 (17.64)	148.18 (21.02)	0.43	0.10	5.35 (5.40)	3.38 (10.73)	0.157
BMI	31.49 (7.57)	30.73 (6.88)	32.48 (8.33)	0.14	32.20 (11.67)	33.13 (13.81)	31.01 (8.04)	0.22	0.51	2.40 (9.79)	–1.47 (1.31)	**<0.001**
BMI %	98.20 (1.98)	98.13 (2.22)	98.29 (1.63)	0.59	97.62 (3.04)	97.91 (2.67)	97.25 (3.43)	0.18	0.58	–0.22 (2.11)	–1.04 (1.97)	**0.011**

Data are presented as mean (SD). BMI = body mass index, BMI% = BMI percentile for age and sex, NR = non-responders, R = responders.

**Table 2 nutrients-14-01500-t002:** Changes in Health Behaviors from Initial to Last Visit.

Dietary Categories	BMI-NR	BMI-R	Treatment Effect (BMI = R-NR)	*p*-Value
Second helpings	–0.74	–1.03	–0.29	0.151
Portion size	–0.52	–0.75	–0.23	0.086
Sugar-sweetened beverages	–0.78	–1.26	–0.47	**0.019**
Processed food	–0.38	–0.87	–0.49	**0.002**
Sweets	–0.54	–1.20	–0.66	**<0.001**
Unhealthy snacks	–0.56	–1.03	–0.46	**0.009**
Skipping meals	–0.04	–0.14	–0.09	0.346
Meals eaten at school	–0.34	–0.53	–0.19	0.222
Condiment use	–0.44	–0.64	–0.19	0.093
Fruits and vegetables	–0.51	–0.75	–0.24	0.065
Whole grains	–0.37	–0.53	–0.16	0.095

Data are presented as mean (SD). BMI = body mass index, NR = non-responders, R = responders.

**Table 3 nutrients-14-01500-t003:** Prevalence of Metabolic dysfunction.

	Initial Visit				Last Visit				Interval Change			
	Total	BMI- NR	BMI- R	*p* R-NR	Total	BMI-NR	BMI- R	*p* R-NR	*p* Total	Mean ∆, BMI-NR	Mean ∆, BMI-R	*p* R-NR
HbA1c	5.27 (0.38)	5.25 (0.33)	5.29 (0.44)	0.51	5.25 (0.35)	5.23 (0.32)	5.28 (0.39)	0.38	0.98	–0.01 (0.19)	–0.01 (0.22)	0.824
Glucose, mg/dL	87.10 (6.47)	87.14 (6.46)	87.04 (6.54)	0.92	88.07 (6.22)	88.91 (6.11)	86.95 (46.26)	0.07	0.33	1.76 (6.31)	–0.09 (2.39)	**0.021**
ALT, U/L	30.76 (27.69)	30.00 (30.91)	31.75 (22.95)	0.68	26.36 (18.09)	26.73 (20.23)	25.87 (14.93)	0.76	0.14	–3.27 (17.44)	–5.88 (16.39)	0.338
Triglycerides, mg/dL	121.86 (65.90)	120.50 (60.65)	123.67 (72.74)	0.76	116.52 (70.55)	127.47 (82.77)	101.91 (46.54)	**0.01**	0.001	6.97 (57.14)	–21.75 (53.11)	**0.001**
Total Chol, mg/dL	158.81 (29.03)	154.27 (30.09)	164.87 (26.57)	**0.02**	154.91 (26.98)	153.30 (28.22)	157.04 (25.29)	0.38	0.008	–0.97 (15.52)	–7.83 (20.87)	**0.023**
LDL-c, mg/dL	89.71 (24.88)	86.38 (26.44)	94.14 (22.05)	**0.05**	86.99 (22.85)	84.42 (23.67)	90.41 (21.40)	0.10	0.28	–1.96 (15.65)	–3.74 (17.11)	0.493
HDL-c, mg/dL	44.84 (8.56)	43.51 (8.75)	46.62 (8.03)	**0.02**	44.53 (9.01)	43.32 (9.46)	46.16 (8.14)	**0.04**	0.64	–0.20 (5.09)	–0.46 (4.56)	0.730

Data are presented as mean (SD). ALT = alanine aminotransferase, BMI = body mass index, Total Chol = total cholesterol, HbA1c = glycated hemoglobin, HDL-c = high-density lipoprotein cholesterol, LDL-c = low-density lipoprotein cholesterol, NR = non-responders, R = responders. **∆** = delta.

## Data Availability

We did not generate large data sets that should be reported into public repository.

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
