# Peer review of "Dietary Counseling Aimed at Reducing Sugar Intake Yields the Greatest Improvement in Management of Weight and Metabolic Dysfunction in Children with Obesity"

_nutrients, 2022, doi:10.3390/nu14071500_

Round 1
Reviewer 1 Report
Comments:
The research article “Dietary Counseling Aimed at Reducing Sugar Intake Yields the Greatest Improvement in Management of Weight and Metabolic Dysfunction in Obese Children” indicate cut down consumption of foods with high sugar content to reduce metabolic disorder in pediatric population. The study is presented in systematic way and well written. Addressing the few minor issues will improve the quality of manuscript:
Comment 1: Understanding the maternal obesity is important as link between obese mother and child obesity is playing critical role in pediatric obesity. There is need to focus on this aspect in the manuscript in discussion section.
Comment 2: Gestational diabetes increase the risk of childhood obesity. Add few lines related with gestational diabetes and risk of pediatric obesity in discussion section.
Comment 3: No change observed in serum ALT level in BMI-R and BMI-NR groups. What is status of Serum AST and serum ALP level between these groups?
Author Response
Dear reviewers,
Thank you for your constructive feedback. Please see the attached response.

Reviewer 2 Report
Abstract, line 33, 'fruits and vegetables and consumption of whole grains' overlooks significant advances in our understanding of the top killer worldwide, overweight.
Preface - Alcohol, and drug misuse were considered weaknesses in self-discipline and control before the disease concept of substance abuse circa 1985. The same holds for nicotine misuse in 1991. The transition to brief medical detoxification and major lifestyle transitioning proceeded well until laboratories and pharmaceutical companies looked for and dispersed a magic pill or procedure to treat substance misuse, bypassing the most critical part, lifestyle transitioning.
The terms fruits and vegetables and consumption of whole grains in this paper reflect understandable omissions in the new clinical management of overweight. They overlook at a minimum:
Zhang Q, Ames JM, Smith RD, Baynes JW, Metz TO. A Perspective
on the Maillard Reaction and the Analysis of Protein
Glycation by Mass Spectrometry: Probing the Pathogenesis of
Chronic Disease. J Proteome Res. 2009 Feb 6; 8(2): 754–69.
Cocores JA, Gold MS. The Salted Food Addiction Hypothesis
may explain overeating and the obesity epidemic. 2009
Dec;73(6):892-9.
Almajwal AM, Alam I, Abdulmeaty M, et al. Intake of dietary
advanced glycation end products influences inflammatory
markers, immune phenotypes, and antiradical capacity of
healthy elderly in a little-studied population. Food Sci Nutr.
2020 Jan 10;8(2):1046-57.
Straub RH, Schradin C. Chronic inflammatory systemic diseases:
An evolutionary trade-off between acutely beneficial but
chronically harmful programs. Evol Med Public Health. 2016
(1): 37-51.
Gistera A, Hansson GK. The immunology of atherosclerosis.
Nat Rev Nephrol. 2017 Jun;13(6):368-380.
Khansari N, Shakiba Y, Mahmoudi M. Chronic inflammation
and oxidative stress as a major cause of age-related diseases
and cancer. Recent Pat Inflamm Allergy Drug Discov. 2009
Jan; 3(1):73.80.
Daley CA, Abbott A, Doyle PS, et al. A review of fatty acid
profiles and antioxidant content in grass-fed and grain-fed beef.
Nutr J. 2010 Mar 10; 9:10.
Perkins TN, Oczypok EA, Dutz RE, et al. (2019) The receptor for advanced glycation end products is a critical mediator of type 2 cytokine signaling in the lungs. J Allergy Clin Immunol 144:796-808. [Example of Type I hypersensitivity (Ex-TIH)]
Nettis E, Distaso M, Saitta S, et al. (2017) Involvement of new oxidative stress markers in chronic spontaneous uticaria. PostepyDermatolAlergol34:448-452. [Ex-TIH]
Angelopoulou E, Paudel YN, Piperi C. (2020) Unraveling the Role of Receptor for Advanced Glycation End Products (RAGE) and Its Ligands in Myasthenia Gravis. ACS ChemNeurosci 11:663-673. [Ex-TIIH]
Abel M, Ritthaler U, Zhang Y, et al. (1995) Expression of receptors for advanced glycosylated end-products in renal disease. Nephrol Dial Transplant 10:1662-1667. [Ex-TIIIH]
Martens HA, Nienhuis HLA, Gross S, et al. (2012) Receptor for advanced glycation end products (RAGE) polymorphisms are associated with systemic lupus erythematosus and disease severity in lupus nephritis. Lupus. 21:959-968. [Ex-TIIIH]
Hong JY, Kim MJ, Hong JK, et al. (2020) In vivo quantitative analysis of advanced glycation end products in atopic dermatitis-Possible culprit for the comorbidities? Exp Dermatol 29:1012-1016. [Ex-TIVH]
Gao Y, Wake H, Morioka Y, et al. (2017) Phagocytosis of Advanced Glycation End Products (AGEs) in Macrophages Induces Cell Apoptosis. Oxid Med Cell Longev8419035. [Ex-TIVH]
Straub RH (2014) TRPV1, TRPA1, and TRPM8 channels in inflammation, energy redirection, water retention: role in chronic inflammatory diseases with an evolutionary perspective. J Mol Med (Berl) 92:925-37. [Another example of body volume expansion not due to calories in and out]
The terms sugar (cane or other sugar in beverages), fruits (fructose sugar), vegetables, and whole grains reflect inaccurate, old, and antiquated assumptions about nutrition, metabolism, and overweight that can be updated to make this an essential contribution to our understanding of nutrition and metabolism.
Beverage sugar is an opiate/dopamine agonist and bleached melanoidin that triggers pan-cellular and systemic inflammation (references below).
A percentage of the fruits consumed by each subject may need approximation regarding the Maillard reaction: Heated by the sun or body (i.e., conventional, or organic apple), intermediate Maillard antigens (i.e., baked apple pie), melanoidins (i.e., apple chips). The raw organic apple is an ani-inflammatory, the pie leads to moderate systemic inflammation or overweight, and the chips obesity.
Similarly, vegetables may need to be categorized accordingly to make more accurate statistical and clinical extrapolations.
Whole grains are essentially non-existent in the human diet. Wheat, barley, rice, and the like dried at antigen-producing temperatures, often heated a second time during pulverization into flour. Oxidized a third time by bleaching and a forth time by baking, sautéing, or frying.
Organic corn eaten fresh in the field is highly anti-inflammatory and gradually becomes a more powerful antigen as the processing temperature increases from canning to the grill.
Flour-containing baked goods, gluten-free or not, can have a higher glycemic index than granular sugar.
Oats dried at high temperatures similarly gradate to stronger antigens.
The term whole grains are most often plant-based acrylamide-rich coated and contain acrylamide/melanoidin-type antigens.
Like alcohol, drugs, and nicotine, it is time to move away from the incorrect concept that excess calories from processed sugar or grains, fats, and proteins lead to weight gain in sedentary and stressed people.
Animal and plant-based foods are denatured when heated above sunlight and body temperatures, transforming nutrients into antigens. Processed sugar, foods, sweets (flour powder or ‘sugar’, no longer whole grains), and unhealthy snacks (a combination of the preceding) consist of Maillard intermediate and end-products: melanoidins, advanced glycation end-products, and advanced lipoxidation end-products. Maillard intermediate and end-products are often antigens that trigger Type I, II, III, and IV hypersensitivity immune responses. And systemic inflammation is misperceived as overweight from excess calories, stagnation, and emotional and systemic oxidative stress.
This article can evolve into a significant paper by converting the data to reflect the percentage of the overheated Maillard-food types and their impact on BMI and waist-height ratio.
Likely, more than half did not slim down because they only reduced one of the many kinds of Maillard coated and containing foods and beverages. The paper remains good because, clinically, the new opiate/dopamine agonist and immuno-inflammation or weight expansion model starts with one type, sugar, plus a fresh-frozen boiled as directed lima bean or gandules and organic first pressed olive oil in tinted glass Maillard-chelating shake. It is too difficult for counselors to sell the reduction of all highly addictive Maillard foods. It is less complicated but not easy to counsel towards reducing or stopping one highly opiate/dopamine addictive and immuno-activating toxic food such as beverage sugar.
Author Response

(The authors gave the same response as above.)
